# Genome-wide association meta-analysis identifies 29 new acne susceptibility loci

Brittany L. Mitchell [1,2,21], Jake R. Saklatvala [3,21], Nick Dand [3,4], Fiona A. Hagenbeek [5,6], Xin Li [7,8], Josine L. Min [9,10], Laurent Thomas [11,12,13], Meike Bartels [5,6], Jouke Jan Hottenga [5,6], Michelle K. Lupton[1], Dorret I. Boomsma[5,6], Xianjun Dong [14,15], Kristian Hveem[16,17,18], Mari Løset [16,19], Nicholas G. Martin [1], Jonathan N. Barker[20], Jiali Han[7,8], Catherine H. Smith[20], Miguel E. Rentería [1,2 ✉] & Michael A. Simpson [3 ✉]

Acne vulgaris is a highly heritable skin disorder that primarily impacts facial skin. Severely inflamed lesions may leave permanent scars that have been associated with long-term psychosocial consequences. Here, we perform a GWAS meta-analysis comprising 20,165 individuals with acne from nine independent European ancestry cohorts. We identify 29 novel genome-wide significant loci and replicate 14 of the 17 previously identified risk loci, bringing the total number of reported acne risk loci to 46. Using fine-mapping and eQTL colocalisation approaches, we identify putative causal genes at several acne susceptibility loci that have previously been implicated in Mendelian hair and skin disorders, including pustular psoriasis. We identify shared genetic aetiology between acne, hormone levels, hormone-sensitive cancers and psychiatric traits. Finally, we show that a polygenic risk score calculated from our results explains up to 5.6% of the variance in acne liability in an independent cohort.

[1] Department of Genetics and Computational Biology, QIMR Berghofer Medical Research Institute, Brisbane, Australia. [2] School of Biomedical Sciences, Faculty of Health, Queensland University of Technology (QUT), Brisbane, Australia. [3] Department of Medical and Molecular Genetics, King's College London, London, UK. [4] Health Data Research UK, London, UK. [5] Department of Biological Psychology, Vrije Universiteit Amsterdam, Amsterdam, The Netherlands. [6] Amsterdam Public Health research institute, Amsterdam, The Netherlands. [7] Department of Epidemiology, Indiana University Richard M. Fairbanks School of Public Health, Indianapolis, US. [8] Indiana University Melvin and Bren Simon Comprehensive Cancer Center, Indianapolis, US. [9] MRC Integrative Epidemiology Unit, University of Bristol, Bristol, UK. [10] Population Health Sciences, Bristol Medical School, University of Bristol, Bristol, UK. [11] Department of Clinical and Molecular Medicine, Norwegian University of Science and Technology, Trondheim, Norway. [12] K. G. Jebsen Center for Genetic Epidemiology, Department of Public Health and Nursing, Faculty of Medicine and Health, Norwegian University of Science and Technology, Trondheim, Norway. [13] BioCore - Bioinformatics Core Facility, Norwegian University of Science and Technology, Trondheim, Norway. [14] Genomics and Bioinformatics Hub, Brigham and Women's Hospital, Boston, MA, USA. [15] Department of Neurology, Brigham and Women's Hospital, Boston, MA, USA. [16] K.G. Jebsen Center for Genetic Epidemiology, Department of Public Health and Nursing, NTNU, Norwegian University of Science and Technology, Trondheim, Norway. [17] HUNT Research Centre, Department of Public Health and Nursing, Norwegian University of Science and Technology, Levanger, Norway. [18] Levanger Hospital, Nord-Trøndelag Hospital Trust, Levanger, Norway. [19] Department of Dermatology, Clinic of Orthopaedy, Rheumatology and Dermatology, St. Olavs Hospital, Trondheim University Hospital, Trondheim, Norway. [20] St John's Institute of Dermatology, Faculty of Life Sciences & Medicine, King's College London, London, UK. [21] These authors contributed equally: Brittany L. Mitchell, Jake R. Saklatvala. ✉email: miguel.renteria@qimrberghofer.edu.au; michael.simpson@kcl.ac.uk

Acne vulgaris is a common skin disorder that results from inflammation of the pilosebaceous unit leading to the characteristic comedones, papules, pustules, nodules and cysts. Lesions are generally restricted to the face, neck, chest and back, and onset typically occurs during puberty. Prevalence estimates of acne vary substantially; it is estimated that more than 85% of teenagers are affected to some degree, and up to 8% have been reported with severe disease[1], making acne the most prevalent skin disease worldwide[2]. According to the Global Burden of Diseases, Injuries, and Risk Factors Study (GBD)[3] acne was estimated to be responsible for nearly 5 million disability-adjusted life years (DALYs) globally in 2019, of which the majority occurred in those aged 15–49 years. This is greater than other chronic inflammatory conditions such as psoriasis or rheumatoid arthritis. Severe acne often persists into adulthood, and scar formation is more prevalent in this severe adult population. Acne is associated with impaired quality of life, including lower rates of employment in patients with acne, and reduced school and work performance[4,5], as well as a substantial impact on mental health that correlates with clinical severity[4,6]. Individuals with acne have elevated rates of depression and higher rates of mental health illness than those without acne[7], with rates of suicidal ideation and other mental health disorders being up to three times higher in adolescents with severe acne than their peers with little or no acne[4]. Several studies report increased clinical depression, anxiety and hospitalisation rates due to mental health disorders in adults and adolescents with acne, with the magnitude of the effect larger in adults[8,9].

Despite considerable recent advances in new treatments for other inflammatory skin diseases, including psoriasis and atopic dermatitis, there is a substantial unmet medical need in the treatment of acne. Early and effective intervention strategies are often necessary to avoid irreversible scarring in severe acne. Treatment typically consists of topical and systemic agents that suppress the microbiome repertoire or the activity of sebaceous glands, while other treatments include hormonal treatment and phototherapy. The most effective agent to treat acne is isotretinoin, which may induce remission through its effect on epidermal differentiation, but its side effect profile, which includes dry skin, lips (cheilitis) and mucous membranes, muscle aches, itching of the skin and headaches[10], restricts use to only those with severe disease. Isotretinoin is also a powerful teratogen whose use during pregnancy is restricted due to its association with severe and life-threatening birth defects. Better-tolerated acne treatments are required and there is a particular need for effective options without the risk of teratogenicity.

The genetic contribution to acne susceptibility has been demonstrated in several twin studies, with heritability consistently estimated around 80%[11–15]. Recent molecular genetic studies identified 17 genomic loci harbouring alleles associated with the disease—15 loci with a reported effect in European populations[16,17] and two in a Han Chinese population[18]. Functional characterisation of these genetic association signals has implicated a series of causal genes whose genetic perturbation impacts the development and maintenance of the hair follicle and wound healing.

To further characterise the genetic architecture of acne vulgaris and identify additional genomic loci contributing to the disease susceptibility, we have performed a meta-analysis of genome-wide association studies (GWAS) of acne undertaken in nine independent cohorts that in total comprise 615,396 study participants (20,165 cases and 595,231 controls). Next, we combine fine-mapping and genome-wide analytical approaches to gain insights into the underlying genes and pathways through which the associated loci contribute to disease susceptibility, and the relationship between the genetic architecture of acne and other traits.

## Results

**Meta-analysis of acne GWAS.** We conducted case-control GWAS of acne in fourteen datasets from nine independent European ancestry cohorts (Supplementary Note). Ascertainment of acne case status varied across the individual cohorts from clinical diagnoses of acne vulgaris by a dermatologist to self-reported disease (Supplementary Data 1). The resulting meta-analysis of the fourteen GWAS datasets demonstrated moderate inflation of test statistics ($\lambda GC = 1.14$, Supplementary Fig. 1), though the LD-score regression (LDSC) intercept (1.02) indicated that the inflation is driven by trait polygenicity rather than confounding bias. We identify genome-wide significant ($P < 5 \times 10^{-8}$) associations at 43 loci (Fig. 1, Table 1, Supplementary Figs. 4, 5), comprising 46 independent genetic variants associated with acne, with LD-based clumping indicating there are two independent ($r^2 < 0.001$) genome-wide significant associations at three of the previously established loci (1q41, 5q11.2, 11q13.1).

We observe association at 14 of the 17 previously reported acne susceptibility loci and 29 novel acne susceptibility loci (Fig. 1, Table 1). We fail to replicate the two acne risk loci previously reported in the Han Chinese population, 1q24.2 and 11p11.2, with neither of the reported lead variants reaching statistical significance in our meta-analysis (rs7531806 at 1q24.2: $P = 0.336$; rs747650 at 11p11.2: $P = 0.0785$) and minimal evidence of effect in our separate cohorts (Supplementary Data 2; Supplementary Fig. 6). The third previously reported acne susceptibility locus for which we do not observe an association at genome-wide significance is located at 2q14.2. However, we observe evidence of a sub-genome-wide significant effect of the lead variant, rs1092479, consistent with the previously reported magnitude and direction on acne risk (OR = 1.06, $P = 9.84 \times 10^{-6}$; Supplementary Fig. 6).

**Identification of causal variants and genes.** Statistical fine-mapping of each association signal revealed two loci at which a single variant was identified as the putative causal variant with a posterior probability >0.95 (Table 1). This includes rs1256580 at the *TGFB2* locus and rs260643 as a candidate causal variant in the novel acne susceptibility locus at 2q12.3. rs260643 is located in a transcription factor binding site in a region of open chromatin within intron 5 of *EDAR*. Rare protein-coding variants in *EDAR* have been demonstrated to cause both recessive and dominant forms of ectodermal dysplasia (OMIM:224900, OMIM:129490), and acne risk allele of rs260643 itself has been reported to be associated with hair curl in a GWAS in Japanese women[19,20]. The range of phenotypes that result from functional variation at this locus closely resembles the known acne susceptibility locus harbouring *WNT10A*, at which genetic variation influences acne risk, hair curl, male pattern baldness and Mendelian forms of ectodermal dysplasia.

To identify additional putative causal genes at acne susceptibility loci, we used a combination of approaches: (a) colocalisation of the acne association signals with skin eQTL signals (Supplementary Data 3), (b) the identification of protein-coding variation within the 95% credible set (Supplementary Data 4) and (c) bioinformatics approaches to identify groups of genes with a related biological function that are located in the proximity of multiple associated loci (Supplementary Fig. 2, Supplementary Data 5, 6). This analytical approach strongly highlights the importance of genes implicated in cellular adhesion and motility. The support for these cellular processes arises from several putative causal genes previously implicated at established acne susceptibility loci, including *LAMC2*, *TGFB2*, *WNT10A*, *LGR6*, *FGF2* and *GLI2* but also highlights the potential consistency of

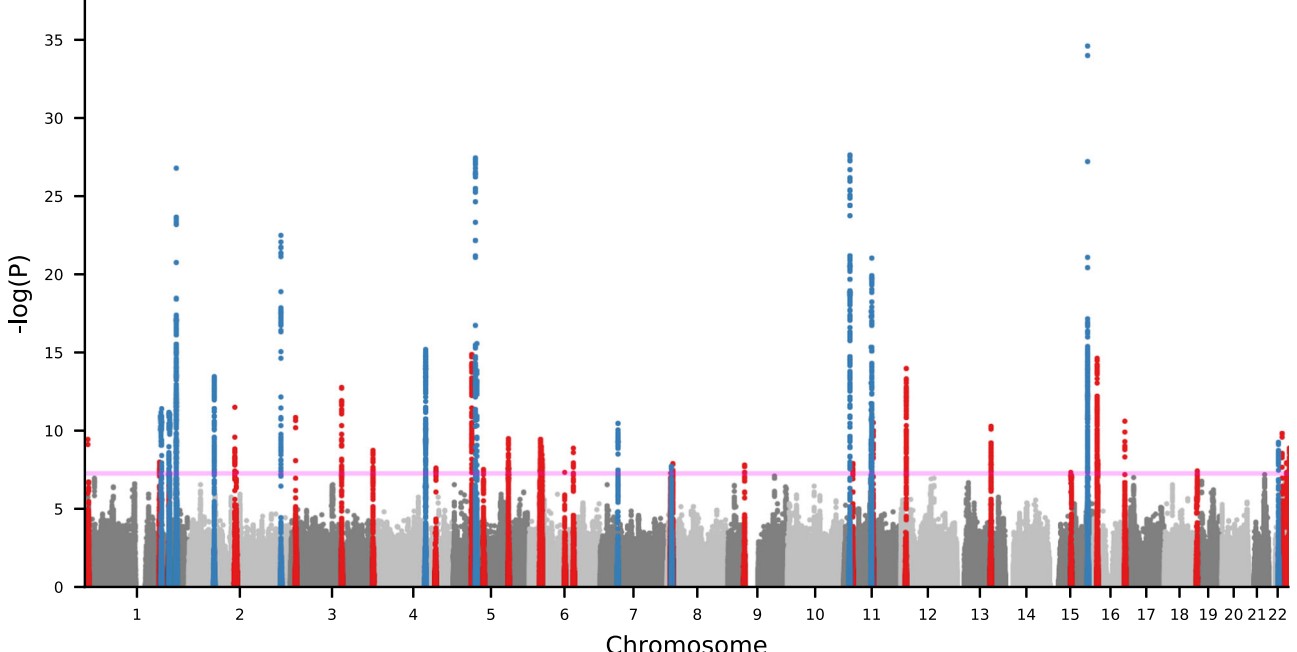

**Fig. 1 Manhattan plot showing genome-wide significant loci associated with acne (20,165 cases, 595,231 controls).** Axes contain a point for each genetic variant passing QC ordered by chromosome and base position on the x-axis, with $-\log_{10}(P\text{-value})$ of association (two-sided Z-test, not adjusted for multiple comparisons) plotted on the y-axis. Blue indicates variants within previously established loci and red indicates variants within novel loci. Pink line indicates genome-wide significance threshold ($P = 5 \times 10^{-8}$).

biological processes involved in acne pathogenesis with the identification of other members of these gene sets located near new acne association signals.

*EDNRA* is highlighted as a potential causal gene at the acne risk locus at 4q31.22. *EDNRA* was strongly implicated as the causal gene at this locus through the DEPICT analysis ($P = 1.4 \times 10^{-4}$; Supplementary Data 5). The same genetic variant that influences acne risk is also associated with *EDNRA* expression in both sun-exposed and non-sun-exposed skin ($PP_{SE} = 0.98$, $PP_{NSE} = 0.98$; Supplementary Data 3), with the acne risk allele associated with a decrease in *EDNRA* expression. *EDNRA* encodes an endothelin receptor in which rare missense variants cause mandibulofacial dysostosis (OMIM:616367), where patterning defects of the hair follicle lead to alopecia.

At 3q21.1, there is further evidence supporting the importance of cell-cell adhesion processes in the skin in acne susceptibility through the identification of *CSTA* as the putative causal gene at this locus. *CSTA* encodes Cystatin A, a protease inhibitor with an established role in cell-cell adhesion. The acne susceptibility signal at 3q21.1 shares a common causal variant with skin eQTLs for *CSTA* with high probability ($PP_{SE} = 0.95$, $PP_{NSE} = 0.98$), with the acne risk allele lowering expression of *CSTA*. Homozygous loss-of-function of *CSTA* causes a peeling skin syndrome that results from extensive hyperkeratosis (OMIM:607936). There is also evidence of a potential shared biological mechanism with pustular psoriasis at 2q13. The acne susceptibility association signal colocalises with *IL36RN* eQTLs in both sun-exposed and non-sun-exposed skin ($PP_{SE} = 0.63$, $PP_{NSE} = 0.62$). The acne risk allele is associated with decreased *IL36RN* expression, which is directionally consistent with the rare putative loss of function missense variants in *IL36RN* underlying pustular forms of psoriasis (generalised pustular psoriasis, acrodermatitis continua of Hallopeau and palmoplantar pustulosis)[21,22].

**Genetic correlations and causal relationships of acne with other traits.** Assuming a population prevalence of 30% for acne,

the genome-wide significant acne risk loci explain an estimated 6.01% of the variance in acne liability. However, estimation of heritability explained by all common SNPs, i.e., the SNP-based heritability, indicates that 22.95% (s.e. = 0.02) of the variance in acne liability is explained by common genetic variation across the genome. We utilised this extensive polygenicity to examine the genetic correlation and potential causal relationship between acne and a series of 935 human diseases and traits, finding 45 traits with statistically significant genetic correlations (Supplementary Data 7). As has been previously observed, there is evidence of genetic correlation between acne and Crohn's Disease (rg = 0.19, s.e. = 0.07) (Fig. 2a). We also observe evidence of shared genetic architecture with disease traits that are phenotypically associated with acne; this includes breast cancer (rg = 0.16, s.e. = 0.05) and psychiatric disorders such as schizophrenia (rg = 0.18, s.e. = 0.06) and bipolar disorder (rg = 0.12, s.e. = 0.05). There is also evidence of asymmetry in the observed genetic correlation between acne and endogenous testosterone and bilirubin levels, breast cancer, joint pain and headaches (Fig. 2b, Supplementary Data 7).

**Polygenic prediction of acne risk.** We evaluated the potential for a polygenic risk score (PRS) that estimates an individual's genetic liability of acne to predict the phenotypic expression in an independent cohort of 2,058 people for whom the history of acne had been evaluated by questionnaire. The polygenicity of acne susceptibility was further assessed by comparing a PRS constructed with the lead variants at genome-wide significant loci ($P < 5 \times 10^{-8}$) and a PRS leveraging genome-wide effects across all SNPs with the SBayesR algorithm[23]. Whilst both scores were strongly associated with acne (defined as moderate or severe acne) in this independent cohort ($P_{GWS\ threshold} = 1.05 \times 10^{-6}$, $P_{SBayesR} = 1.68 \times 10^{-12}$), the cumulative risk derived from genome-wide significant association signals explained 2.8% (s.e. = 1.32%) of variance in acne liability compared to 5.6% (s.e. = 1.79) using SBayesR (Supplementary Fig. 3). In line with our prediction results, individuals reporting moderate or severe

**Table 1 Meta-analysis genome-wide significant loci.**

| SNP | CHR | BP | Band | Established/Novel | RAF | RA | PA | OR (95% CI) | P-value | 95% CS SNPs | Lead SNP PP | Putative causal gene(s) |
|---|---|---|---|---|---|---|---|---|---|---|---|---|
| rs8029268 | 1 | 8207579 | 1p36.23 | Novel | 0.0504 | C | G | 1.21 (1.14–1.28) | $3.56 \times 10^{-10}$ | 2 | 0.6492 | ERRFI1[c] |
| rs404818 | 1 | 179254597 | 1q25.2 | Novel | 0.2531 | T | C | 1.08 (1.05–1.11) | $9.76 \times 10^{-9}$ | 49 | 0.0841 | SOAT1[c] |
| rs513398 | 1 | 183163127 | 1q25.3 | Established[18] | 0.56 | A | G | 1.09 (1.06–1.11) | $3.69 \times 10^{-12}$ | 275 | 0.0187 | LAMC2[b,c], LAMC1[d] |
| rs296522 | 1 | 200881940 | 1q32.1 | Novel | 0.8192 | C | T | 1.09 (1.06–1.12) | $3.10 \times 10^{-8}$ | 29 | 0.1391 | INAVA (C1orf106)[c,d] |
| rs6658708 | 1 | 202322113 | 1q32.1 | Established[18] | 0.527 | T | G | 1.08 (1.06–1.11) | $6.19 \times 10^{-12}$ | 109 | 0.0291 | LGR6[b], PPP1R12B[c] |
| rs1256580 | 1 | 219199380 | 1q41 | Established[17] | 0.1517 | C | G | 1.19 (1.15–1.23) | $1.49 \times 10^{-27}$ | 1 | 0.9962 | TGFB2[b], LYPLAL1[c] |
| rs6684734 | 1 | 219631744 | 1q41 | Established[17] | 0.3993 | G | A | 1.08 (1.05–1.1) | $5.06 \times 10^{-10}$ | * | * | * |
| rs2901000 | 2 | 60501216 | 2p16.1 | Established[18] | 0.4285 | A | G | 1.09 (1.07–1.12) | $3.13 \times 10^{-14}$ | 32 | 0.0881 | BCL11A[b,c] |
| rs260643 | 2 | 109539653 | 2q12.3 | Novel | 0.0998 | A | G | 1.15 (1.07–1.12) | $2.93 \times 10^{-12}$ | 1 | 0.9581 | EDAR[c] |
| rs6735739 | 2 | 113609886 | 2q13 | Novel | 0.334 | T | C | 1.07 (1.04–1.1) | $4.39 \times 10^{-8}$ | 9 | 0.3052 | IL36RN[a], IL1B[c] |
| rs74333950 | 2 | 219746292 | 2q35 | Established[18] | 0.8603 | T | G | 1.2 (1.16–1.24) | $3.03 \times 10^{-23}$ | 5 | 0.5707 | WNT10A[b,c] |
| rs3773364 | 3 | 12189968 | 3p25.2 | Novel | 0.1574 | G | A | 1.12 (1.08–1.15) | $1.27 \times 10^{-11}$ | 3 | 0.563 | TIMP4[c] |
| rs17265703 | 3 | 122048644 | 3q21.1 | Novel | 0.1521 | G | A | 1.13 (1.08–1.15) | $1.57 \times 10^{-13}$ | 18 | 0.2569 | CSTA[a,c] |
| rs34381158 | 3 | 196946647 | 3q29 | Novel | 0.326 | G | A | 1.08 (1.09–1.17) | $1.67 \times 10^{-9}$ | 44 | 0.1754 | DLG1[a,c] |
| rs13104688 | 4 | 124179519 | 4q28.1 | Established[18] | 0.3426 | G | T | 1.11 (1.05–1.11) | $6.28 \times 10^{-16}$ | 82 | 0.0488 | FGF2[b], SPRY1[c] |
| rs6842241 | 4 | 148400819 | 4q31.22 | Novel | 0.8622 | C | A | 1.1 (1.08–1.13) | $2.22 \times 10^{-8}$ | 20 | 0.0753 | EDNRA[a,c] |
| rs16874036 | 5 | 44366552 | 5p12 | Novel | 0.6926 | G | A | 1.11 (1.06–1.14) | $1.22 \times 10^{-15}$ | 25 | 0.1212 | FGF10[c] |
| rs629725 | 5 | 52631067 | 5q11.2 | Established[17] | 0.3393 | T | C | 1.15 (1.08–1.14) | $3.50 \times 10^{-28}$ | 22 | 0.1488 | FST[b,c] |
| rs158343 | 5 | 55624905 | 5q11.2 | Established[18] | 0.2006 | G | C | 1.13 (1.12–1.17) | $2.50 \times 10^{-16}$ | 3 | 0.9332 | ANKRD55[c] |
| rs455660 | 5 | 55816888 | 5q11.2 | Established[18] | 0.8215 | C | T | 1.09 (1.1–1.17) | $1.42 \times 10^{-8}$ | * | * | * |
| rs258887 | 5 | 72277724 | 5q13.2 | Novel | 0.5439 | A | C | 1.07 (1.06–1.12) | $2.78 \times 10^{-8}$ | 40 | 0.1211 | FCHO2[a,c] |
| rs11242109 | 5 | 131677047 | 5q31.1 | Novel | 0.4719 | T | G | 1.08 (1.04–1.09) | $2.92 \times 10^{-10}$ | 127 | 0.1503 | SLC22A5[a,c], PDLIM4[a], SLC22A4[d] |
| rs169262 | 6 | 27770890 | 6p22.1 | Novel | 0.8091 | C | T | 1.1 (1.05–1.1) | $3.24 \times 10^{-10}$ | 626 | 0.0239 | * |
| rs17692425 | 6 | 85582396 | 6q14.3 | Novel | 0.3443 | C | T | 1.07 (1.07–1.14) | $4.16 \times 10^{-8}$ | 15 | 0.678 | TBX18[c] |
| rs9398069 | 6 | 106601165 | 6q21 | Novel | 0.6127 | T | C | 1.08 (1.05–1.1) | $1.29 \times 10^{-9}$ | 9 | 0.4395 | PRDM1[c] |
| rs9639838 | 7 | 40873916 | 7p14.1 | Established[18] | 0.2087 | T | C | 1.1 (1.05–1.1) | $3.27 \times 10^{-11}$ | 41 | 0.1889 | SUGCT[c] |
| rs2945230 | 8 | 8109936 | 8p23.1 | Novel | 0.5252 | G | A | 1.07 (1.07–1.13) | $4.61 \times 10^{-8}$ | 134 | 0.1319 | PRAG1[c] |

**Table 1 (continued)**

| SNP | CHR | BP | Band | Established/Novel | RAF | RA | PA | OR (95% CI) | P-value | 95% CS SNPs | Lead SNP PP | Putative causal gene(s) |
|---|---|---|---|---|---|---|---|---|---|---|---|---|
| rs919555 | 8 | 1061876 | 8p23.1 | Established[18] | 0.5352 | C | A | 1.07 (1.04-1.09) | $1.73 \times 10^{-8}$ | 64 | 0.1138 | SOX7[c] |
| rs17803958 | 8 | 13231742 | 8p22 | Novel | 0.07166 | T | C | 1.14 (1.09-1.19) | $1.13 \times 10^{-8}$ | 34 | 0.1619 | C8orf48[c] |
| rs4878737 | 9 | 38020544 | 9p13.2 | Novel | 0.2568 | T | G | 1.08 (1.05-1.11) | $1.40 \times 10^{-8}$ | 10 | 0.2135 | SHB[a,c] |
| rs3849154 | 11 | 13124524 | 11p15.2 | Established[18] | 0.3237 | T | G | 1.15 (1.12-1.17) | $2.22 \times 10^{-28}$ | 6 | 0.3589 | RASSF10[c] |
| rs1838055 | 11 | 19871230 | 11p15.1 | Novel | 0.1135 | C | G | 1.11 (1.07-1.15) | $1.13 \times 10^{-8}$ | 18 | 0.0961 | DBX1[c] |
| rs144908022 | 11 | 64892198 | 11q13.1 | Established[17] | 0.01025 | G | A | 1.61 (1.44-1.79) | $2.19 \times 10^{-18}$ | * | * | * |
| rs61744384 | 11 | 65387378 | 11q13.1 | Established[17] | 0.5586 | T | A | 1.12 (1.1-1.15) | $8.58 \times 10^{-22}$ | 29 | 0.3958 | MAP3K11[a,b], OVOL1[b], PCNX3[c] |
| rs10896460 | 11 | 69073216 | 11q13.3 | Novel | 0.8125 | T | G | 1.11 (1.08-1.14) | $2.80 \times 10^{-11}$ | 10 | 0.3247 | MYEOV[c] |
| rs7312010 | 12 | 12612436 | 12p13.2 | Novel | 0.4064 | A | G | 1.1 (1.07-1.12) | $9.85 \times 10^{-15}$ | 40 | 0.2544 | BORCS5[c] |
| rs1342583 | 13 | 81441795 | 13q31.1 | Novel | 0.5695 | A | G | 1.08 (1.06-1.11) | $4.85 \times 10^{-11}$ | 20 | 0.3461 | SPRY2[c] |
| rs8042919 | 15 | 50878630 | 15q21.2 | Novel | 0.1109 | A | G | 1.11 (1.07-1.15) | $4.48 \times 10^{-8}$ | 60 | 0.0271 | SPPL2A[a], USP50[c], TRPM7[d] |
| rs34560261 | 15 | 90734426 | 15q26.1 | Established[18] | 0.8345 | C | T | 1.25 (1.21-1.3) | $2.51 \times 10^{-35}$ | 2 | 0.8487 | SEMA4B[a,b,c] |
| rs7194305 | 16 | 11099707 | 16p13.13 | Novel | 0.584 | A | G | 1.1 (1.07-1.13) | $2.18 \times 10^{-15}$ | 19 | 0.1628 | CLEC16A[c] |
| rs72803831 | 16 | 77515430 | 16q23.1 | Novel | 0.8875 | G | A | 1.15 (1.1-1.19) | $2.40 \times 10^{-11}$ | 5 | 0.6983 | ADAMTS18[c] |
| rs12373373 | 18 | 77984345 | 18q23 | Novel | 0.3977 | T | C | 1.08 (1.05-1.11) | $3.49 \times 10^{-8}$ | 81 | 0.1086 | PARD6G[c] |
| rs2070475 | 22 | 24891355 | 22q11.23 | Established[18] | 0.1641 | T | A | 1.11 (1.07-1.15) | $5.40 \times 10^{-10}$ | 16 | 0.2138 | SPECC1L[a,b], UPB1[c] |
| rs135025 | 22 | 33202478 | 22q12.3 | Novel | 0.4414 | A | G | 1.08 (1.05-1.1) | $1.42 \times 10^{-10}$ | 6 | 0.2566 | TIMP3[c] |
| rs738409 | 22 | 44324727 | 22q13.31 | Novel | 0.7782 | C | G | 1.09 (1.06-1.12) | $1.00 \times 10^{-8}$ | 24 | 0.2281 | PNPLA3[c,d] |
| rs28470568 | 22 | 50335839 | 22q13.33 | Novel | 0.1448 | T | G | 1.11 (1.08-1.15) | $1.19 \times 10^{-9}$ | 118 | 0.0771 | CRELD2[a], ALG12[a,d], PIM3[c] |

*RAF* risk allele frequency (GNOMAD, North-western Europeans), *RA* risk allele, *PA* protective allele, *OR* odds ratio, *CI* confidence intervals, *P-value* two-sided Z-test (not adjusted for multiple comparisons), *ABF* approximate Bayes factor, *CS* credible set.
Putative causal genes are labelled as following:
[a]Implicated through eQTL colocalization evidence PP > 0.5 in both sun-exposed and not sun-exposed skin.
[b]Gene implicated in Petridis et al.
[c]Nearest protein-coding gene to SNP.
[d]Protein-altering SNP in 95% credible set (Supplementary Data 4).
*Variant is second independent signal at locus.

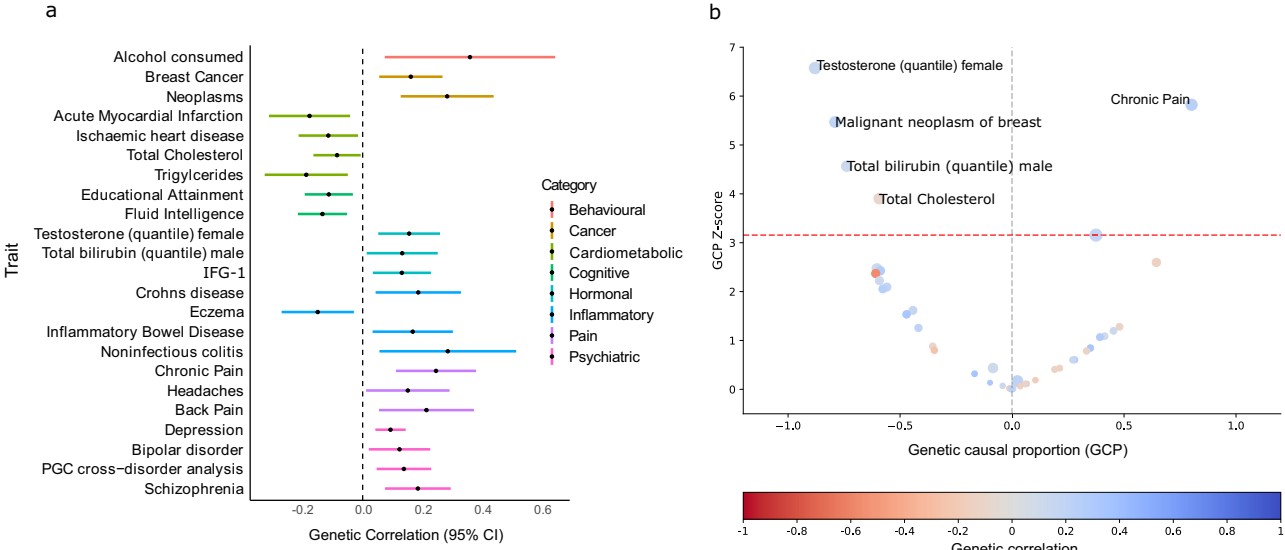

**Fig. 2 Genetic correlation and latent causal variable analysis between acne and other complex traits.** All analyses were conducted using GWAS summary statistic data from 935 complex traits in the CTG-VL platform. **a** Black circles represent point estimates of LD score-based genetic correlations. Error bars indicate 95% confidence intervals. **b** Colour bar indicates strength and direction of genetic correlation where red indicates a negative correlation and blue a positive correlation. Red line indicates significance threshold for multiple testing (FDR < 5%). CI confidence intervals, GCP Genetic causal proportion.

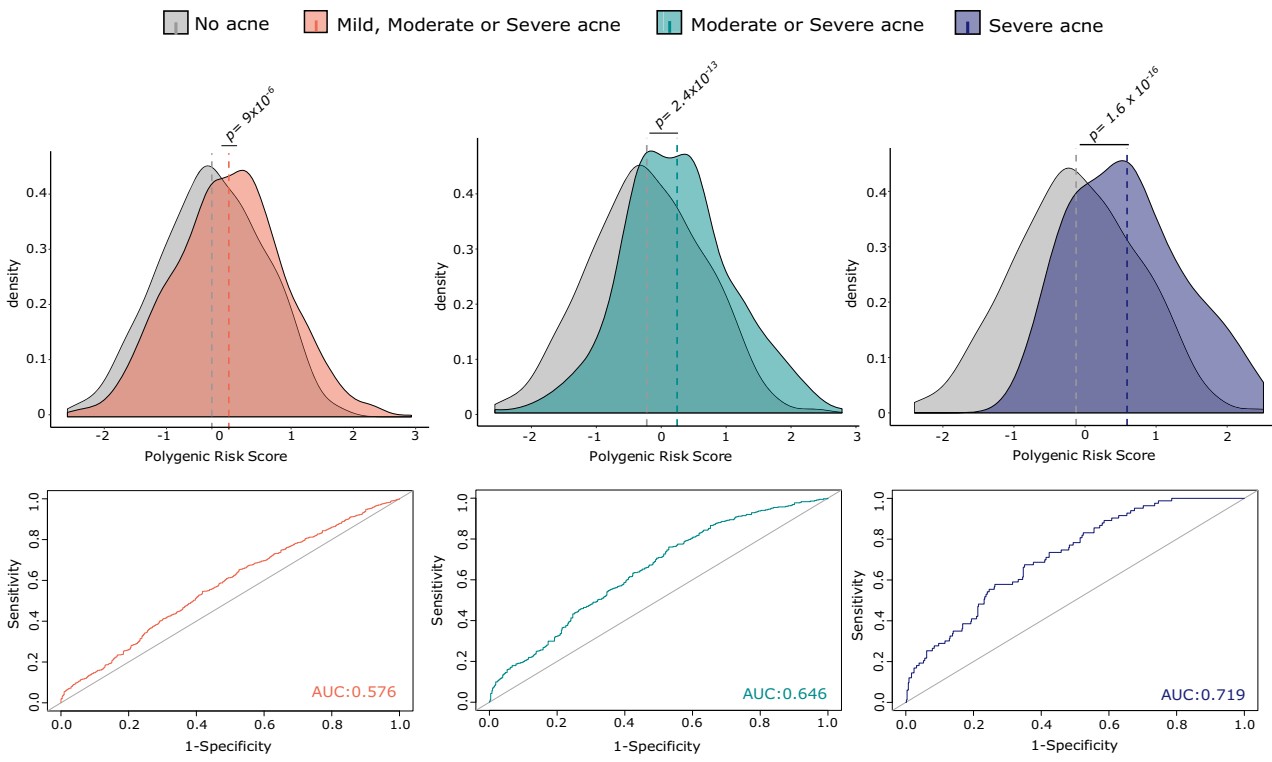

**Fig. 3 Acne PRS analysis and prediction.** Top panels show the difference in distributions of mean acne PRSs between individuals that report no acne (grey) to those that report having mild, moderate or severe acne (orange), those that report only moderate or severe acne (green) or those that report only having severe acne (purple). *P*-values were calculated using a two-sided Student's *t*-test. Bottom panels depict the corresponding increase in predictive ability of the acne PRS in these different case groups using receiver operating characteristic (ROC) curves. AUC Area under the curve.

acne had significantly higher mean acne PRSs than those that reported no acne ($P_{\text{Moderate-none}} = 2.4 \times 10^{-13}$; $P_{\text{Severe-none}} = 1.6 \times 10^{-16}$; Fig. 3). Similarly, when assessing the ability of acne PRS to predict increasingly strict definitions of acne, we find that the acne PRS has the greatest predictive ability in individuals with severe acne (AUC 0.7) (Fig. 3).

## Discussion

Acne is a complex human trait, with heritability estimates of up to 80% reported consistently in twin studies[11,13]. The current study represents an approximate fourfold increase in the number of cases compared to the previous meta-analysis[17]. Notably, we do not observe evidence of association in the current meta-

analysis, or any of the contributing studies, at either of the two acne risk loci observed in the Han Chinese population[18], highlighting potential differences in the genetic architecture of acne between different ethnic populations. This warrants further investigation in studies of diverse ancestry.

At several of the newly identified acne risk loci, we find evidence of the importance of the structure and morphology of the hair follicle in disease susceptibility. There are strong parallels between the phenotypic consequences of the allelic series of variation at the *EDAR* locus and the previously implicated risk locus at *WNT10A*. At both of these loci, acne susceptibility, hair morphology and ectodermal dysplasia all result from genetic variation that impacts the function of the respective genes. The presence of other Mendelian skin disease genes at other risk loci also provides key insight into the Mendelian mechanisms that contribute to acne susceptibility. This includes evidence for the development and maintenance of the pilosebaceous unit, indicated by the presence of several ectodermal dysplasia genes at acne risk loci, and also neutrophilic inflammation evidence by the presence of an acne susceptibility association signal at the IL36RN locus at which rare loss of function alleles have been associated with a series of pustular skin phenotypes[21,22,24].

In addition to identifying specific loci contributing to acne susceptibility, our results demonstrate substantial polygenicity beyond the genome-wide significant association signals. In addition to the estimated 6% of the variance in acne liability explained by the 46 independent genome-wide significant variants identified here, a further 17% of acne liability is estimated to be explained by the polygenic tail. The polygenic architecture enabled the evaluation of genetic relationships with other traits that share underlying biological pathways, including immune-mediated disorders including Crohn's disease and evidence of shared genetic aetiology with other traits including a series of mental health disorders and breast cancer. The epidemiological association between acne and mental health has been well documented. However, the direction of causality remains unclear; several observational studies indicate that damaged self-esteem resulting from acne development may mediate the association between severe acne and internalising psychopathology[25,26] and depression, anxiety, and emotional lability have been associated with isotretinoin treatment[27]. In contrast there is also evidence that stress, poor self-care and drug treatments for mental health disorders can cause acne[25,28,29]. Elevated rates of breast cancer have been previously highlighted in females with a history of acne[30] and the shared influence of genetics between acne and breast cancer highlights the potential importance of endogenous hormone regulation, which is a key component of breast cancer risk and is further supported by a putative causal relationship between testosterone levels and acne. We also identify shared genetic architecture between acne and chronic pain, specifically joint pain and headache. Notably, there appears to be asymmetry in the genetic correlation, consistent with loci contributing to acne susceptibility being causal for chronic pain.

The current study capitalised on acne diagnoses ascertained by multiple different approaches, from clinical diagnosis to self-reported disease with varying criteria for case and control definition. This heterogeneity in cohort definitions may introduce bias into the effect size estimates, with estimates from the clinically ascertained cohorts typically larger than those observed in the cohorts where acne was self-reported or ascertained from electronic health records. In addition, the exclusion of mild acne cases from some of the self-reported cohorts may lead to inflated effect size estimates. Nevertheless, the acne PRS defined from the meta-analysis is strongly associated with self-reported acne history across mild, moderate and severe groups. Consistent with a

liability threshold model of acne risk, the generated acne PRSs were associated with disease severity, with the best prediction among the severe group. Whilst further validation is needed, a robust genetic predictor of acne risk may have utility in identifying individuals at the highest risk of acne and intervene with prophylactic skin care regimes to minimise follicular occlusion before bacterial colonisation and extensive inflammation is established to reduce disease severity and scarring. The current study sought to establish genetic risk factors that are independent of sex and age, however, further investigation to define specific genetic loci with sex or age-specific effects will improve our mechanistic understanding of the disease and has the potential to improve polygenic prediction by modelling of dimorphic effects.

In summary, the results of the current study represent a transformational increase in our understanding of the genetic basis of acne. Interrogation of these loci further illustrates the shared biology processes with other skin and hair traits and interrogation of genome-wide genetic liability identified shared genetic aetiology with other common diseases. Our results highlight the substantial influence on genetic risk harboured by other, as yet undiscovered loci and motivate future studies to both identify additional risk loci and establish the biological processes through which genetic risk is mediated.

## Methods

**Cohort details.** Data were collected from 9 collaborating centres, with some centres providing independent GWAS for more than one cohort, creating a total of 14 independent datasets. Acne vulgaris definitions varied between the cohorts and consisted of clinical assessment, electronic health record coding and self-reported diagnoses of acne, resulting in a final sample size of 20,165 cases and 595,231 controls (Supplementary Data 1). Detailed information regarding informed consent, ethical approval, recruitment, genotyping, QC and GWAS in each cohort is described in Supplementary Note 1.

**Meta-analysis.** We conducted an inverse-variance-weighted meta-analysis of 615,396 individuals (20,165 cases and 595,231 controls) from each of the cohorts described above (14 datasets) using METAL[31]. All variants were aligned to positions on human build GRCh37 (hg19), and variants with MAF < 1% and an imputation accuracy score <0.7 were excluded prior to the meta-analysis. Only variants present in at least 9 of the 14 datasets were included in the final meta-analysis, resulting in 7,072,770 variants.

**Locus definition/LD clumping.** Linkage disequilibrium (LD) clumping was used to identify the positions of loci containing acne-associated variants. Clumping of the results from the meta-analysis was conducted using Plink v1.9[32] with the following parameters: a *P*-value cut-off of $5 \times 10^{-8}$, 1 Mb distance between variants and $r^2 < 0.001$ for variants within the genomic distance cut-off, using the linkage disequilibrium structure of the European ancestry subset of the 1000 Genomes Project as a reference panel. Due to the complexity of the major histocompatibility complex region (chr6:26-34 Mb), only the most significant variant in this locus has been reported.

**Fine-mapping.** An approximate Bayes factor was calculated from the effect size and standard error of each variant in each associated locus (lead SNP ± 1 Mb), using the approach defined by Wakefield[33], assuming a prior variance on the log odds ratios of 0.04 (Eq. (1)). The resulting Bayes factors were then re-scaled to reflect the posterior probability for each variant being causal, and 95% credible sets were defined as the minimal set of variants whose combined posterior probabilities sum to ≥0.95. Where there were multiple independently segregating SNPs within the flanking 1 Mb regions, fine-mapping was only performed once, using the SNP with lowest *P*-value as lead SNP.

$$\text{ABF} = \sqrt{\frac{V + W}{V}} \exp\left(-\frac{z^2}{2}\frac{W}{(V + W)}\right) \qquad (1)$$

$\sqrt{V}$ is the standard error of the maximum likelihood estimate; $z^2$ is Wald statistic ($\beta^2/V$); $W$ is prior variance on the log odds ratios.

**eQTL colocalisation.** We examined the colocalisation between acne association signals and skin cis-eQTLs from the GTEx (The Genotype-Tissue Expression project) consortium. Candidate skin eQTLs were defined as any variant located within an acne risk locus (±1 Mb from lead SNP) that was also associated with variation in the expression of a nearby gene ($P < 1 \times 10^{-4}$). A Bayesian test for colocalisation between the acne association signal and the skin eQTL signal was

performed using a set of variants that overlapped between the two studies using the R package coloc[34], with a prior probability of colocalisation defined as $P$: $10^{-5}$. A posterior probability exceeding 50% was used as evidence of colocalisation.

**Identification of coding SNPs in loci 95% credible sets**. For each locus (excluding HLA), all variants present in the 95% credible sets were annotated using Ensembl Variant Effect Predictor[35]. Both risk and protective alleles were queried. The HLA locus was excluded.

**Gene and gene-set annotation**. To gain insights into the underlying genes and pathways that explain our associations, we used DEPICT (https://github.com/perslab/depict) to prioritise causal genes and identify gene-sets and tissue types in which these genes are enriched[36]. DEPICT uses 14,461 reconstituted gene-sets, in each of which a z-score of association is calculated for each gene across the genome. DEPICT prioritises genes at associated loci by calculating how highly correlated they are with other genes in that gene-set. We used a $P$-value threshold of $1 \times 10^{-5}$ to define loci associated with acne. We corrected for multiple testing using Benjamini-Hochberg's False Discovery Rate (FDR < 5%). Further annotation of links, including membership of gene families, co-existence within curated pathways, co-expression across tissues and experimentally determined interactions was performed between genes located in acne susceptibility loci was performed using Ensembl[37] and STRING databases[38] databases. The low-confidence threshold (0.150) was used to define links between genes in STRING.

**Heritability, genetic correlations and latent causal variable analysis**. We used LD-Score (LDSC) regression[39] and the HapMap3 reference panel to estimate the total SNP-based heritability ($h^2_{SNP}$) of the acne meta-analysis. We also estimated the variance explained by the genome-wide significant SNPs alone using R package Mangrove (https://CRAN.R-project.org/package=Mangrove). For both analyses, we assumed a population prevalence of 30%. We used the Complex Traits Genetics Virtual Lab (CTG-VL, https://genoma.io/) to calculate genetic correlations between our acne meta-analysis and 935 complex behavioural and disease traits and screen these traits for a potential causal association with acne using the Latent Causal Variable (LCV) method[40]. LCV does not directly test for causality but instead estimates a genetic causal proportion (GCP) parameter that mediates an association between two traits; a GCP of 0 indicates no genetic causal association, and a GCP of 1 indicates full genetic causality. GCP values lower than 1 but greater than 0 indicate partial causality. In this study, we considered a GCP > 0.7 as an indication of a causal relationship. We applied a multiple testing correction (FDR < 5%) to determine statistical significance.

**PRS analysis**. We leveraged our meta-analysis results to build acne PRS and assess their predictive ability in an independent sample. We used acne data collected in the Prospective Imaging of Aging Study (PISA) at QIMR Berghofer as our target sample. PISA currently consists of ~3000 genotyped individuals who have completed extensive behavioural, psychological and medical questionnaires and cognitive testing and brain imaging[41]. Participants were asked about the presence and severity of acne as a teenager to which they could answer None, Mild, Moderate or Severe. We conducted two regression analyses: first, using all participants with cases coded on an ordinal scale and second, using only individuals that answered Moderate or Severe included as cases in our analysis ($N = 527$), and individuals that reported not having acne considered controls ($N = 645$). Individuals were genotyped as described in the Supplementary Note. To avoid bias due to potential sample overlap between PISA and the other QIMR cohorts included in the meta-analysis, we used genotype information to exclude all individuals with an IBD > 0.125—which equates to third-degree relatives.

We calculated PRS using both SBayesR v2.03 and the traditional clumping and thresholding (C+T) methods. SBayesR is a Bayesian method that assumes that SNP effects are drawn from a mixture of four zero-mean normal distributions with different variances[23]. This method re-scales the GWAS SNP effects with many SNPs assumed to have an effect size of zero. SBayesR has been shown to outperform other PRS methods in the prediction of complex traits. For the LD reference, we used the same sparse LD matrix as in Lloyd-Jones et al.[23], where the LD matrix was built based on the HapMap3 SNPs of randomly selected and unrelated 50,000 UK Biobank individuals. The posterior SNP effects were used to generate PRS for each individual using the --score function in PLINK. PRS were calculated using the C+T method across eight different SNP $P$-value significance thresholds: $P < 5 \times 10^{-8}$, $P < 1 \times 10^{-5}$, $P < 0.001$, $P < 0.01$, $P < 0.05$, $P < 0.1$, $P < 0.5$, $P < 1$. For each individual, at each threshold, an acne PRS was calculated by multiplying the dosage score and effect size for each SNP, and then these values were summed across all loci. For more detail on C+T PRS calculation see Mitchell et al.[42].

For each PRS, a mixed model regression was conducted with the acne PRS as a predictor variable while accounting for sex, age and the first ten genetic principal components (to account for residual population stratification) as fixed effects; relatedness among individuals was accounted for as a random effect with a genetic relatedness matrix, as implemented in GCTA 1.91.7[43]. A partial $R^2$ was used to estimate the variance explained by the PRS. The predictive ability of the PRS was further evaluated using receiver operating characteristic (ROC) curves using ROC curves in R v3.6.1 using the pRoc package[44]. Significance values were calculated using a two-tailed Student's $t$-test.

**Reporting summary**. Further information on research design is available in the Nature Research Reporting Summary linked to this article.

## Data availability

The GWAS summary statistics are available in the GWAS catalogue (www.ebi.ac.uk/gwas) with accession number GCST90092000.

StringDB version 11.5 was used, accessed at: https://stringdb-static.org/download/protein.links.detailed.v11.5/9606.protein.links.detailed.v11.5.txt.gz

GTEx v8 (EUR) was used, accessed at: https://console.cloud.google.com/storage/browser/gtex-resources

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

## Acknowledgements

We thank the participants who donated their time, life experiences and DNA to this research, and the clinical and scientific teams that worked with them. We acknowledge support from the National Institute for Health Research (NIHR), through the NIHR Biomedical Research Centre based at Guy's and St Thomas' NHS Foundation Trust and King's College London. Health Data Research UK (MR/S003126/1). Acknowledgments for each cohort that contributed to this meta-analysis can be found in the supplementary information.

## Author contributions

M.E.R. and M.A.S. contributed equally to the study design with help from B.L.M., J.R.S., J.N.B., C.H.S. and N.G.M. M.E.R. and M.A.S. contributed equally to the supervision of the study. B.L.M. and J.R.S. performed the meta-analysis and all subsequent post-GWAS analyses. J.R.S. and N.D. performed UK Biobank GWAS with help from C.H.S. to define acne phenotype. J.L.M. performed ALSPAC GWAS. L.T., M.L. and K.H. performed HUNT GWAS. X.D. performed the GWAS on the Mass General Brigham Biobank. J.H. and X.L. performed the GWAS of the Harvard datasets. M.K.L. provided access to the PISA dataset used for the PRS analysis. N.G.M. provided QIMR cohort data and B.L.M. ran the GWAS. F.A.H., J.J.H., M.B. and D.I.B. were responsible for data collection, genotyping and GWAS analysis of NTR data. All co-authors contributed to drafting and provided relevant input for this manuscript.

## Competing interests

The authors declare no competing interests.
