## [Peer Review File · Nature Communications]

Genome-wide association meta-analysis identifies 29 new acne susceptibility lociReviewers' Comments:

Reviewer #1:

Remarks to the Author:

Mitchell et al. report on the results of a large European ancestry meta-analysis on acne vulgaris comprising more than 20,000 cases and almost 600,000 controls. The report advances the field by expanding the number of known acne risk loci to 46. The authors further apply typical bioinformatics approaches to interpret the association findings and to identify putative candidate genes and pathways. The study seems straightforward and the methodology sound. I have the following comments:

1. A gender bias for acne risk loci has been previously reported. Did the authors test for sex-specific effects in their study? What is the gender distribution in their meta-analysis cohort?
2. The supplementary data include a detailed but lengthy description of all cohorts. I'd encourage the authors to present the key attributes of each study (including genotyping conditions, imputing strategy and reference panel, age and gender distribution) in table format (e.g. add to Supplementary table1).
3. The authors use teen and adult cohorts for their analysis. Do their data allow any conclusions on the shared or differing genetic basis of teenage-only vs persistent acne? How does clinical vs. self-reported diagnosis impact their association findings?
3. Please add references for previously reported loci in table 1
4. Can the authors add a few sentences on the limitations of their present study and potential future directions in acne genetic research?
5. The phrase "the allelic spectrum of phenotypic consequences" does not sound coherent. Can the authors rephrase
6. The authors report candidate genes at different risk loci to be involved in similar biological processes. Can the authors include a figure to illustrate these functional interactions between acne risk loci. Did they run any additional test for (functional) interaction between risk variants/loci?

Reviewer #2:

Remarks to the Author:

Mitchell et al. describe results from the largest GWAS meta-analysis of acne performed to date. They analyzed 20,165 individuals with acne from nine independent cohorts and identified 29 novel genome-wide significant loci and replicated 14 of the 17 previously identified risk loci, bringing the total number of reported acne risk loci to 46. Implicated genes included those implicated in Mendelian hair and skin disorders, including pustular psoriasis. The genome-wide significant acne risk loci are proposed to explain an estimated 6.01% of the variance in acne liability and 22.95% of the variance in acne liability is explained by common genetic variation across the genome. A polygenic risk score calculated from their associated SNPs explained up to 5.6% of the variance in acne liability in an independent cohort.

The study was well done and state of the art statistical approaches were used. The manuscript and figures/tables are clear. I have only minor comments:

There is redundancy when results and discussion are compared (for example the discussion of acne and depression, breast cancer etc (in this reviewer's mind this is also a stretch in terms of correlation)). If these are to be included, please only include in the discussion. Moreover, some of the discussion also has redundancy within it (for example reiterating the number of acne loci identified).

In the tables provide the genome build.

Were any of the risk alleles coding SNPs, or in high LD with coding SNPs?

In Table 1 some SNPs lie in plausible genes yet they weren't annotated – e.g. rs135025 lies within TIMP3 which is a perfectly reasonable candidate. Moreover, a second risk locus harbors TIMP4. Can the authors perhaps comment on this.

Reviewer #1 (Remarks to the Author):

Mitchell et al. report on the results of a large European ancestry meta-analysis on acne vulgaris comprising more than 20,000 cases and almost 600,000 controls. The report advances the field by expanding the number of known acne risk loci to 46. The authors further apply typical bioinformatics approaches to interpret the association findings and to identify putative candidate genes and pathways. The study seems straightforward and the methodology sound. I have the following comments:

1. A gender bias for acne risk loci has been previously reported. Did the authors test for sex-specific effects in their study? What is the gender distribution in their meta-analysis cohort?

We thank the reviewer for highlighting the importance of potential sex differences in the pathogenesis of acne. The meta-analysis performed in the current study was not designed to specifically address gender specific effects and has been performed across cohorts comprising both males and females. We have added details of the sex distributions of each of the cohorts to Supplementary table 1.

2. The supplementary data include a detailed but lengthy description of all cohorts. I'd encourage the authors to present the key attributes of each study (including genotyping conditions, imputing strategy and reference panel, age and gender distribution) in table format (e.g. add to Supplementary table1).

We thank the reviewer for this suggestion, which we agree will make the summaries of each cohort more easily digestible and comparable. As suggested, we have added the relevant information to Supplementary table 1.

3. The authors use teen and adult cohorts for their analysis. Do their data allow any conclusions on the shared or differing genetic basis of teenage-only vs persistent acne? How does clinical vs. self-reported diagnosis impact their association findings?

Identifying genetic variation that contributes to the persistence of acne into adulthood is a key objective of acne research. Unfortunately, the ascertainment of the datasets in the current study does not provide the ability to robustly address this question. The majority of the cohorts are focused on a teen acne phenotype. However, the primary challenge to undertaking a robust 'teen vs adult' analysis in the current dataset reflects that diagnosis of acne in three of the four 'adult' acne datasets (FinnGen, HUNT and UKB) are primarily drawn from electronic health records and will therefore likely represent a mixture of transient teenage and acne that persists into adulthood. We also note that the names QIMR cohorts ("adult" and "teen") were slightly misleading because these were descriptors of the cohort at ascertainment, whereas the questionnaires used in the adult cohort asked about teenage acne. We have updated the cohort names to "self-report" (formerly the adult cohort) and "clinical" (formerly the teen cohort).

3. Please add references for previously reported loci in table 1

We have added in references for the previously reported loci as requested.

4. Can the authors add a few sentences on the limitations of their present study and potential future directions in acne genetic research?

We appreciate the request to outline future directions for acne research and have added further details to the discussion describing the limitations of the current study in terms of the heterogeneity of diagnostic approaches between cohorts and the resulting heterogeneity, the limitations in the ability to address sex and age effects and highlighted how these may be addressed in future studies.

5. The phrase “the allelic spectrum of phenotypic consequences” does not sound coherent. Can the authors rephrase

We agree, this was not the most elegant phraseology, we have edited this section for clarity.

6. The authors report candidate genes at different risk loci to be involved in similar biological processes. Can the authors include a figure to illustrate these functional interactions between acne risk loci. Did they run any additional test for (functional) interaction between risk variants/loci?

We thank the reviewer for this suggestion and have added a supplementary figure (Supplementary figure 2) that illustrates the functional links between genes implicated across different acne susceptibility loci.

Reviewer #2 (Remarks to the Author):

Mitchell et al. describe results from the largest GWAS meta-analysis of acne performed to date. They analyzed 20,165 individuals with acne from nine independent cohorts and identified 29 novel genome-wide significant loci and replicated 14 of the 17 previously identified risk loci, bringing the total number of reported acne risk loci to 46. Implicated genes included those implicated in Mendelian hair and skin disorders, including pustular psoriasis. The genome-wide significant acne risk loci are proposed to explain an estimated 6.01% of the variance in acne liability and 22.95% of the variance in acne liability is explained by common genetic variation across the genome. A polygenic risk score calculated from their associated SNPs explained up to 5.6% of the variance in acne liability in an independent cohort.

The study was well done and state of the art statistical approaches were used. The manuscript and figures/tables are clear. I have only minor comments:

There is redundancy when results and discussion are compared (for example the discussion of acne and depression, breast cancer etc (in this reviewer's mind this is also a stretch in terms of correlation)). If these are to be included, please only include in the discussion. Moreover, some of the discussion also has redundancy within it (for example reiterating the number of acne loci identified).

We thank the reviewer for highlighting some elements of redundancy between the results and discussion. We have edited elements of both these sections to minimise repetition, including minimising the description of the findings from the genetic correlation analysis in the results section.

In the tables provide the genome build.

All analysis were performed with reference to GRCh37, we have updated the tables with the details and added a sentence to the methods section to ensure this is clear.

Were any of the risk alleles coding SNPs, or in high LD with coding SNPs?

We have updated table 1 to include details of 6 genes for which a protein coding variant is a member of the 95% credible set of putative causal variants for an acne association signal.

In Table 1 some SNPs lie in plausible genes yet they weren't annotated – e.g. rs135025 lies within TIMP3 which is a perfectly reasonable candidate. Moreover, a second risk locus harbors TIMP4. Can the authors perhaps comment on this.

We thank the reviewer for highlighting this candidate gene and its absence from table 1. We had restricted the annotation of genes in table 1 to genes that had convincing evidence of being the causal gene (either being implicated in previously acne GWAS, through colocalisation with skin eQTLs, having coding variation in the credible set or through the DEPICT analysis). However, we agree that this may be an overly conservative approach and have therefore revised the table to additionally include the closest gene (in terms of

distance from the lead variant to the TSS). This includes TIMP3 and TIMP4 and several other candidates.

Reviewers' Comments:

Reviewer #1:

Remarks to the Author:

The authors have adequately addressed all my previous comments.

Reviewer #2:

Remarks to the Author:

The authors have responded to my comments and I don't have any others.